# Learning Incompressible Fluid Dynamics from Scratch - Towards Fast, Differentiable Fluid Models that Generalize

**Nils Wandel**
Department of Computer Science
University of Bonn
wandeln@cs.uni-bonn.de

**Michael Weinmann**
Department of Computer Science
University of Bonn
mw@cs.uni-bonn.de

**Reinhard Klein**
Department of Computer Science
University of Bonn
rk@cs.uni-bonn.de

## Abstract

Fast and stable fluid simulations are an essential prerequisite for applications ranging from computer-generated imagery to computer-aided design in research and development. However, solving the partial differential equations of incompressible fluids is a challenging task and traditional numerical approximation schemes come at high computational costs. Recent deep learning based approaches promise vast speed-ups but do not generalize to new fluid domains, require fluid simulation data for training, or rely on complex pipelines that outsource major parts of the fluid simulation to traditional methods.

In this work, we propose a novel physics-constrained training approach that generalizes to new fluid domains, requires no fluid simulation data, and allows convolutional neural networks to map a fluid state from time-point $t$ to a subsequent state at time $t + dt$ in a single forward pass. This simplifies the pipeline to train and evaluate neural fluid models. After training, the framework yields models that are capable of fast fluid simulations and can handle various fluid phenomena including the Magnus effect and Kármán vortex streets. We present an interactive real-time demo to show the speed and generalization capabilities of our trained models. Moreover, the trained neural networks are efficient differentiable fluid solvers as they offer a differentiable update step to advance the fluid simulation in time. We exploit this fact in a proof-of-concept optimal control experiment. Our models significantly outperform a recent differentiable fluid solver in terms of computational speed and accuracy.

## 1 Introduction

Simulating the behavior of fluids by solving the incompressible Navier-Stokes equations is of great importance for a wide range of applications and accurate as well as fast fluid simulations are a long-standing research goal. On top of simulating the behavior of fluids, several applications such as sensitivity analysis of fluids or gradient-based control algorithms rely on differentiable fluid simulators that allow to propagate gradients throughout the simulation (Holl et al. (2020)).

Recent advances in deep learning aim for fast and accurate fluid simulations but rely on vast datasets and / or do not generalize to new fluid domains. Kim et al. (2019) present a framework to learn parameterized fluid simulations and allow to interpolate efficiently in between such simulations. However, their work does not generalize to new domain geometries that lay outside the training data. Kim & Lee (2020) train a RNN-GAN that produces turbulent flow fields within a pipe domain, but do not show generalization results beyond pipe domains. Xie et al. (2018) introduce a tempoGAN to perform temporally consistent superresolution of smoke simulations. This allows to

produce plausible high-resolution smoke-density fields for arbitrary low-resolution inputs, but our fluid model should output a complete fluid state description consisting of a velocity and a pressure field. Tompson et al. (2017) present how a Helmholtz projection step can be learned to accelerate Eulerian fluid simulations. This method generalizes to new domain geometries, but a particle tracer is needed to deal with the advection term of the Navier-Stokes equations. Furthermore, as Eulerian fluids do not model viscosity, effects like e.g. the Magnus effect or Kármán vortex streets cannot be simulated. Geneva & Zabaras (2020) propose a physics-informed framework to learn the entire update step for the Burgers equations in 1D and 2D, but no generalization results for new domain geometries are demonstrated. All of the aforementioned methods rely on the availability of vast amounts of data from fluid-solvers such as FEniCS, OpenFOAM or Mantaflow. Most of these methods do not generalize well or outsource a major part of the fluid simulation to traditional methods such as low-resolution fluid solvers or a particle tracer.

In this work, we propose a novel unsupervised training framework to learn incompressible fluid dynamics from scratch. It does not require any simulated fluid-data (neither as ground truth data, nor to train an adversarial network, nor to initialize frames for a physics-constrained loss) and generalizes to fluid domains unseen during training. It allows CNNs to learn the entire update-step of mapping a fluid domain from time-point $t$ to $t + dt$ without having to rely on low resolution fluid-solvers or a particle-tracer. In fact, we will demonstrate that a physics-constrained loss function combined with a simple strategy to recycle fluid-data generated by the neural network at training time suffices to teach CNNs fluid dynamics on increasingly realistic statistics of fluid states. This drastically simplifies the training pipeline. Fluid simulations get efficiently unrolled in time by recurrently applying the trained model on a fluid state. Furthermore, the fluid models include viscous friction and handle effects such as the Magnus effect and Kármán vortex streets. On top of that, we show by a gradient-based optimal control example how backpropagation through time can be used to differentiate the fluid simulation. Code and pretrained models are publicly available at `https://github.com/aschethor/Unsupervised_Deep_Learning_of_Incompressible_Fluid_Dynamics/`.

## 2 RELATED WORK

In literature, several different approaches can be found that aim to approximate the dynamics of PDEs in general and fluids in particular with efficient, learning-based surrogate models.

*Lagrangian methods* such as smoothed particle hydrodynamcs (SPH) Gingold & Monaghan (1977) handle fluids from the perspective of many individual particles that move with the velocity field. Following this approach, learning-based methods using regression forests by Ladický et al. (2015), graph neural networks by Mrowca et al. (2018); Li et al. (2019) and continuous convolutions by Ummenhofer et al. (2020) have been developed. In addition, Smooth Particle Networks (SP-Nets) by Schenck & Fox (2018) allow for differentiable fluid simulations within the Lagrangian frame of reference. These Lagrangian methods are particularly suitable when a fluid domain exhibits large, dynamic surfaces (e.g. waves or droplets). However, to simulate the dynamics within a fluid domain accurately, *Eulerian methods*, that treat the Navier-Stokes equations in a fixed frame of reference, are usually better suited.

*Continuous Eulerian methods* allow for mesh-free solutions by mapping domain coordinates (e.g. $x,y,t$) directly onto field values (e.g. velocity $\vec{v}$ / pressure $p$) (Sirignano & Spiliopoulos (2018); Grohs et al. (2018); Khoo et al. (2019)). Recent applications focused on flow through porous media (Zhu & Zabaras (2018); Zhu et al. (2019); Tripathy & Bilionis (2018)), fluid modeling (Yang et al. (2016); Raissi et al. (2018)), turbulence modeling (Geneva & Zabaras (2019); Ling et al. (2016)) and modeling of molecular dynamics (Schöberl et al. (2019)). Training is usually based on physics-constrained loss functions that penalize residuals of the underlying PDEs. Similar to our approach, Raissi et al. (2019) uses vector potentials to obtain continuous divergence-free velocity fields to approximate the incompressible Navier-Stokes equations. Continuous methods return smooth, accurate results and can overcome the curse of dimensionality of discrete techniques in high-dimensional PDEs (Grohs et al. (2018)). However, these networks are trained on a specific domain and cannot generalize to new environments or be used in interactive scenarios.

*Discrete Eulerian methods*, on the other hand, aim to solve the underlying PDEs on a grid and early work dates back to Harlow & Welch (1965) and Stam (1999). Accelerating such traditional works with deep learning techniques is a major field of research and all of the methods mentioned in the introduction fall into this category. Further methods include the approach by Thuerey et al. (2019) to learn solutions of the Reynolds-averaged Navier-Stokes equations for airfoil flows, but requires large amounts of training data and does not generalize beyond airfoil flows. In the work by Um et al. (2020), a correction step is learned that brings solutions of a low-resolution differentiable fluid solver closer to solutions of a high-resolution fluid simulation. However, generalization results for new domain geometries were not presented. The works of Mohan et al. (2020) and Kim et al. (2019) show that vector potentials are suitable to enforce the incompressibility constraint in fluids but do not generalize to new fluid domains beyond their training data.

## 3 METHOD

In this section, we briefly review the incompressible Navier-Stokes equations, which are to be solved by the neural network. Then, we explain how the Helmholtz decomposition can be exploited to ensure incompressibility within the fluid domain. Furthermore, we provide details of our discrete spatio-temporal fluid representation and introduce the fluid model. Afterwards, we formulate a physics-constrained loss function based on residuals of the Navier-Stokes equations and introduce a pressure regularization term for very high Reynolds numbers. Finally, we explain the unsupervised training strategy.

### 3.1 INCOMPRESSIBLE NAVIER-STOKES EQUATIONS

Most fluids can be modeled with the incompressible Navier-Stokes equations - a set of non-linear equations that describe the interplay of a velocity field $\vec{v}$ and a pressure field $p$ within a fluid domain $\Omega$:

$$\nabla \cdot \vec{v} = 0 \qquad \text{incompressibility on } \Omega \qquad (1)$$

$$\rho \dot{\vec{v}} = \rho \left( \frac{\partial \vec{v}}{\partial t} + (\vec{v} \cdot \nabla) \vec{v} \right) = -\nabla p + \mu \Delta \vec{v} + \vec{f} \quad \text{conservation of momentum on } \Omega \qquad (2)$$

Here, $\rho$ describes the fluid density and $\mu$ the viscosity. Equation 1 states that the fluid is incompressible and thus $\vec{v}$ is divergence-free. Equation 2 states that the change in momentum of fluid particles must correspond to the sum of forces that arise from the pressure gradient, viscous friction and external forces. Here, external forces on the fluid (such as e.g. gravity) can be neglected, so we set $\vec{f} = 0$.

These incompressible Navier-Stokes equations shall be solved by a CNN given initial conditions $\vec{v}^0$ and $p^0$ at the beginning of the simulation and Dirichlet boundary conditions which constrain the velocity field at the domain boundary $\partial \Omega$:

$$\vec{v} = \vec{v}_d \qquad \text{Dirichlet boundary condition on } \partial \Omega \qquad (3)$$

### 3.2 HELMHOLTZ DECOMPOSITION

A common method to ensure incompressibility of a fluid (see Equation 1) is to project the flow field onto the divergence-free part of its Helmholtz decomposition. The Helmholtz theorem states that every vector field $\vec{v}$ can be decomposed into a curl-free part ($\nabla q$) and a divergence-free part ($\nabla \times \vec{a}$):

$$\vec{v} = \nabla q + \nabla \times \vec{a} \qquad (4)$$

Note, that $\nabla \times (\nabla q) = \vec{0}$ and $\nabla \cdot (\nabla \times \vec{a}) = 0$. The Helmholtz projection consists of solving the Poisson problem $\nabla \cdot \vec{v} = \Delta q$ for $q$, followed by substracting $\nabla q$ from the original flow field. However, solving the Poisson equation on arbitrary domains comes at high computational costs for classical methods and one has to rely e.g. on conjugate gradient methods to approximate its solution.

Here, we propose a different approach and directly try to learn a vector potential $\vec{a}$ with $\vec{v} = \nabla \times \vec{a}$. This ensures that the network outputs a divergence-free velocity field within the domain $\Omega$ and

automatically solves Equation 1. In this work, we consider 2D fluid simulations, so only the $z$-component of $\vec{a}$, $a_z$, is of interest since $v_z$ and all derivatives with respect to the $z$-axis are zero:

$$\nabla \times \vec{a} = \begin{pmatrix} \partial_y a_z - \partial_z a_y \\ \partial_z a_x - \partial_x a_z \\ \partial_x a_y - \partial_y a_x \end{pmatrix} = \begin{pmatrix} \partial_y a_z \\ -\partial_x a_z \\ 0 \end{pmatrix} = \begin{pmatrix} v_x \\ v_y \\ 0 \end{pmatrix} = \vec{v} \tag{5}$$

### 3.3 Discrete Spatio-temporal Fluid Representation

**Marker-And-Cell (MAC) grid** To solve the Navier-Stokes equations, we represent the relation between $a_z, v_x, v_y, p$ on a 2D staggered marker-and-cell (MAC) grid (see Figure 1a). Therefore, we discretise time and space as follows:

$$\vec{a}(x, y, t) = \begin{pmatrix} 0 \\ 0 \\ (a_z)_{i,j}^t \end{pmatrix} ; \; \vec{v}(x, y, t) = \begin{pmatrix} (v_x)_{i,j}^t \\ (v_y)_{i,j}^t \end{pmatrix} ; \; p(x, y, t) = p_{i,j}^t \tag{6}$$

Obtaining gradient, divergence, Laplace and curl operations on this grid with finite differences is straight forward and can be efficiently implemented with convolutions (see appendix A).

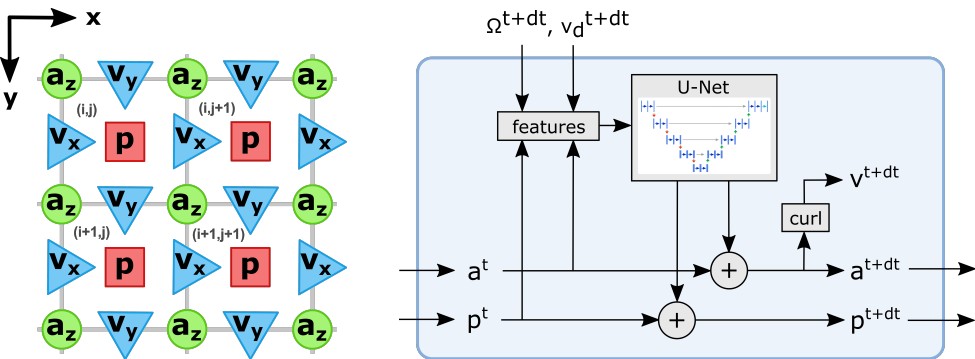

**(a)** Layout of Staggered Marker-And-Cell (MAC) grid in 2D.

**(b)** Diagram of the fluid model. By recurrently applying the model on the fluid state ($p^t$ and $a^t$), we can unroll the fluid simulation in time.

**Figure 1:** MAC grid and diagram of the fluid model.

**Explicit, Implicit, Implicit-Explicit (IMEX) time integration methods** The discretization of the time domain is needed to deal with the time-derivative of the velocity field $\frac{\partial \vec{v}}{\partial t}$ in Equation 2, which becomes:

$$\rho \left( \frac{\vec{v}^{t+dt} - \vec{v}^t}{dt} + \left( \vec{v}^{t'} \cdot \nabla \right) \vec{v}^{t'} \right) = -\nabla p^{t+dt} + \mu \Delta \vec{v}^{t'} + \vec{f} \tag{7}$$

The goal is to take as large as possible timesteps $dt$ while maintaining stable and accurate solutions. Stability and accuracy largely depend on the definition of $v^{t'}$. In literature, choosing $v^{t'} = v^t$ is often referred to as explicit integration methods and frequently leads to unstable behavior. Choosing $v^{t'} = v^{t+dt}$ is usually associated with implicit integration methods and gives stable solutions at the cost of numerical dissipation. Implicit-Explicit (IMEX) methods, which set $v^{t'} = (v^t + v^{t+dt})/2$ are a compromise between both methods and considered to be more accurate but less stable than implicit methods.

### 3.4 FLUID MODEL

We represent the fluid dynamics by a recurrent model that maps the fluid state $p^t, \vec{a}^t$ for timestep $t$ and the domain description $\Omega^{t+dt}, \vec{v}_d^{t+dt}$ to the fluid state $p^{t+dt}, \vec{a}^{t+dt}$ of the next timestep. Here, $p^t$ describes the pressure field and $\vec{a}^t$ describes the vector potential of $\vec{v}^t$. For $t = 0$, we consider initial states $p^0 = 0$ and $\vec{a}^0 = \vec{0}$, however, other initial conditions could be considered as well. $\Omega^{t+dt}$ is a binary mask that contains the domain geometry and is 1 for the fluid domain and 0 everywhere else. For the boundary of the domain, we simply take the inverse of $\Omega$: $\partial\Omega = 1 - \Omega$. $\vec{v}_d^{t+dt}$ represents the Dirichlet boundary conditions and contains a velocity field that must be matched by $\vec{v}^{t+dt}$ at the domain boundaries. Figure 1b shows a diagram of the fluid model. First, $(p^t, \vec{a}^t, \Omega^{t+dt}, \vec{v}_d^{t+dt})$ are taken to derive a slightly more meaningful feature representation that comprises $(p^t, a^t, \nabla \times a^t, \Omega^{t+dt}, \partial\Omega^{t+dt}, \Omega^{t+dt} \cdot \nabla \times a^t, \Omega^{t+dt} \cdot p^t, \partial\Omega^{t+dt} \cdot \vec{v}_d^{t+dt})$. These features can be very efficiently computed with convolutions and are then fed into a U-Net (Ronneberger et al. (2015)) with a reduced number of channels (the exact network configuration can be found in appendix B). The mean of the U-Net output is set to 0 in order to keep $p$ and $\vec{a}$ well defined and prevent drifting offset values. Finally, the output is added to $p^t$ and $\vec{a}^t$ to obtain the updated fluid state $p^{t+dt}$ and $\vec{a}^{t+dt}$.

### 3.5 PHYSICS-CONSTRAINED LOSS FUNCTION

Using the residuals of the Navier-Stokes equations (Equations 1 and 2), we can formulate the following loss terms on $\Omega$ and $\partial\Omega$:

$$L_d = \|\nabla \cdot \vec{v}\|^2 \qquad\qquad \text{divergence loss on } \Omega \qquad (8)$$

$$L_p = \left\|\rho\left(\frac{\partial\vec{v}}{\partial t} + (\vec{v} \cdot \nabla)\vec{v}\right) + \nabla p - \mu\Delta\vec{v} - \vec{f}\right\|^2 \qquad \text{momentum loss on } \Omega \qquad (9)$$

$$L_b = \|\vec{v} - \vec{v}_d\|^2 \qquad\qquad \text{boundary loss on } \partial\Omega \qquad (10)$$

Combining the described loss terms, we obtain the following loss function:

$$L = \alpha L_d + \beta L_p + \gamma L_b \qquad (11)$$

where $\alpha, \beta, \gamma$ are hyperparameters that weight the contributions of the different loss terms. Note that if we use a vector potential $\vec{v} = \nabla \times \vec{a}$, $L_d = 0$ is automatically fulfilled and we can set $\alpha = 0$. This loss function can be computed very efficiently with convolutions in $O(N)$ (where $N$ = number of grid cells), whereas solving the Navier-Stokes equations explicitly would be computationally a lot more expensive. For detailed descriptions regarding the fully discretized loss-function, we refer to appendix A.

### 3.6 PRESSURE REGULARIZATION

For very high Reynolds numbers (see Equation 13) and inviscid flows, training becomes unstable as viscous friction cannot dissipate enough energy out of the system. This leads to unrealistic gradients in $\vec{v}$ and $p$. For such cases, we introduce an additional regularization term for the loss function (11) that can be traded off with $L_p$ to stabilize training:

$$L_r = \|\nabla p\|^2 \qquad (12)$$

The intuition behind this regularization term is, that we want to penalize unrealistically high energies in the pressure field.

### 3.7 TRAINING STRATEGY

Training starts with initializing a pool $\{\Omega_k^0, (v_d)_k^0, (a_z)_k^0, p_k^0\}$ of randomized domains $\Omega_k^0$ and boundary conditions $(v_d)_k^0$ as well as initial conditions for the vector potential and pressure fields that we both set to zero $((a_z)_k^0 = 0$ and $p_k^0 = 0)$. The resolution of our training domains is 100x300 grid cells and example-domains of the training pool are shown in appendix C. Note that our training pool does not rely on any previously simulated fluid-data.

At each training step, a random mini-batch $\{\Omega_k^t, (v_d)_k^t, (a_z)_k^t, p_k^t\}_{\{k \in \text{minibatch}\}}$ is drawn from the pool and fed into the neural network which is designed to predict the velocity $(\vec{v}_k^{t+dt} = \nabla \times \vec{a}_k^{t+dt})$ and pressure $(p_k^{t+dt})$ fields of the next time step. Based on a physics-constrained loss-function (Equation 11), we update the weights of the network using the Adam optimizer (Kingma & Ba (2015)). At the end of each training step, the pool is updated by replacing the old vector potential and pressure fields $(a_z)_k^t, p_k^t$ by the newly predicted ones $(a_z)_k^{t+dt}, p_k^{t+dt}$. This recycling strategy fills the training pool with more and more realistic fluid states as the model becomes better at simulating fluid dynamics.

From time to time, old environments of the training pool are replaced by new randomized environments and the vector potential as well as the pressure fields are reset to 0. This increases the variance of the training pool and helps the neural network to learn "cold starts" from $\vec{0}$-velocity and 0-pressure fields.

Besides the fluid model described above, which we denote as $\vec{a}$-Net in the following, we also trained an ablation model, $\vec{v}$-Net, that directly learns to predict the velocity field without a vector potential. For the implementation of both models, we used the popular machine learning framework Pytorch and trained the models on a NVidia GeForce RTX 2080 Ti. Training converged after about 1 day. The hyperparameters in the loss-function for the $\vec{a}$-Net were $\beta = 1$ and $\gamma = 20$. The reason for choosing a higher weight for the loss term $L_b$ than for $L_p$ was the observation, that errors in $L_b$ can lead to unrealistic flows leaking through boundaries. For the ablation study ($\vec{v}$-Net), we used $\alpha = 100, \beta = 1, \gamma = 0.001$. Here, we had to choose a very high weight for $L_d$ to ensure incompressibility of the fluid, otherwise unrealistic source and sink effects start to appear. For $L_b$, on the other hand, we used a very low weight as the boundary conditions can be trivially learned by the $\vec{v}$-Net. We used these parameter settings for all experiments.

## 4 RESULTS

To evaluate the potential of our method, we assess its ability to reproduce physical effects such as Kármán vortex streets and the Magnus effect. In addition, we demonstrate its generalization capability and real-time performance. Finally, we test the fluid models quantitatively.

### 4.1 QUALITATIVE EVALUATION

**Qualitative analysis of wake dynamics** Qualitative effects in fluid dynamics such as the wake dynamics behind an obstacle are closely related to the Reynolds number. It is a dimensionless quantity defined by:

$$Re = \frac{\rho \|\vec{v}\| D}{\mu} \tag{13}$$

Here, $\rho$ is the fluid density, $\|\vec{v}\|$ is the fluid speed, $D$ is the diameter of the obstacle, and $\mu$ is the viscosity. (We use the units of the grid).

We retrained models for different values of $\mu$ and $\rho$ to compare the fluid behavior for a wide range of Reynolds numbers. Figure 2 shows, that the trained models are able to predict the wake dynamics behind an obstacle in good accordance with qualitative expectations from fluid dynamics. As a rule of thumb, for $Re \ll 1$, the flow becomes time-reversible. This can be noticed in Figure 2a by the symmetry of the flow before and after the obstacle and the nearly constant pressure gradient within the pipe. Starting from $Re \approx 10$, the flow is still laminar but a static wake is forming behind the obstacle (see Figure 2b). For Reynolds numbers $Re >\approx 90$, Kármán vortex streets start to appear (see Figure 2c). A Kármán vortex street consists of clock and counterclockwise spinning vortices that are generated at the obstacle and then start moving in a regularly oscillating pattern with the

flow. For very large Reynolds numbers or inviscid flows, the flow field becomes turbulent, which can be recognized by the irregular patterns behind the obstacle in Fig 2d.

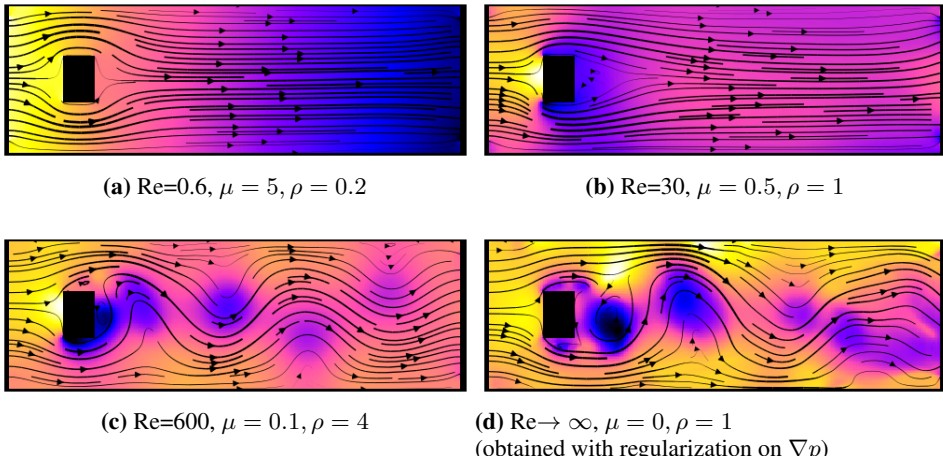

**(a)** Re=0.6, $\mu = 5, \rho = 0.2$    **(b)** Re=30, $\mu = 0.5, \rho = 1$

**(c)** Re=600, $\mu = 0.1, \rho = 4$    **(d)** Re$\to \infty, \mu = 0, \rho = 1$
(obtained with regularization on $\nabla p$)

**Figure 2:** After training, our models are able to show correct wake flow dynamics for a wide range of different Reynolds numbers. ($D = 30, \|\vec{v}\| = 0.5$). Streamlines indicate flow direction, linewidth indicates speed and colors represent the pressure field (blue: low pressure / yellow: high pressure).

**Magnus effect**    The Magnus effect appears when a flow interacts with a rotating body. It is widely known e.g. in sports such as soccer or tennis where spin is used to deflect the path of a ball. The reason for the deflection stems from a low pressure field where the surface of the object moves along flow direction and a high pressure field where the object surface moves against the flow. Figure 3a shows, that our models are able to reproduce the Magnus effect around a rotating cylinder.

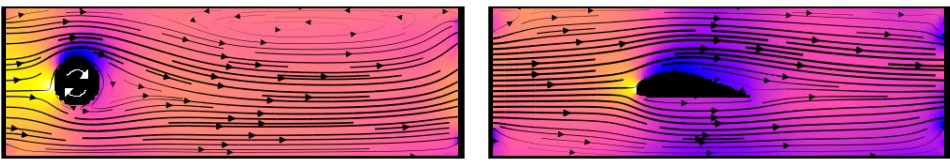

**(a)** Magnus effect on a clock-wise turning cylinder. **(b)** Generalization example: Note that the fluid model has never been confronted with wing-profiles during training.

**Figure 3:** Our models feature the Magnus effect and generalize to new fluid domains. Further examples are presented in appendix D and the video.

**Analysis of generalization capability**    We tested the networks capability to generalize to objects not seen during training. Figure 3b shows the networks capability to meet boundary conditions of an airfoil and return a plausible pressure field that produces lift (see low pressure on top of wing). Note that in contrast to the approach by Thuerey et al. (2019), which learns simplified, time-averaged solutions of the Navier-Stokes equations, our method is able to simulate the full incompressible Navier-Stokes equations for an airfoil without relying on any ground truth data or having seen airfoil-geometries during training. In fact, the network was only trained on simple randomized domains as highlighted in appendix C and Figure 7. Possible reasons for the networks generalization capabilities are:

- During training, the network gets confronted with an infinite number of different flow-fields and randomized domain configurations because the training pool gets updated at every training step. This prevents the network from over-fitting.

- The dynamics of a fluid-particle are mostly determined by its local neighborhood / surrounding particles. This means, the update step for a certain cell on the MAC grid is mostly determined by close / neighboring MAC-grid cells. Since more complicated shapes can be seen locally as a composition of basic shapes (e.g. the front of the wing can be locally regarded as a cylinder), it suffices to train on basic shapes that provide the network with enough examples to generalize to more complicated shapes.

Further generalization examples are provided in appendix D.

**Real-time capability** The fluid simulation can be easily parallelized and takes low computational costs as one time-integration step consists just of a single forward pass through a convolutional neural network. This enables for example interactive real-time simulations. We implemented a demo that allows to interact with a fluid by moving obstacles, rotating spheres and changing the flow speed within a pipe (see video in supplementary material and source code). Our method runs at 250 timesteps per second on a 100x300 grid. In the respective experiments, we used a NVidia GeForce RTX 2080 Ti consuming about 860 MB of GPU memory.

## 4.2 QUANTITATIVE EVALUATION

We compare our method ($\vec{a}$-Net) quantitatively with PhiFlow by Holl et al. (2020). Phiflow is a recent, open source, differentiable fluid simulator based on a MAC grid data structure. Furthermore, we provide an ablation study ($\vec{v}$-Net) that does not make use of the Helmholtz decomposition but directly works on the velocity field $\vec{v}$.

Quantitative comparison of different fluid solvers is challenging, as their performance is highly dependent on factors like the geometry of the domain, fluid parameters such as viscosity or density, flow speed or the timestep of the integrator. As benchmarks for fluid simulations on MAC grids are not yet available, we built a simple toy domain on a 100 x 100 grid which simulates a flow around an obstacle within a pipe (more details are provided in appendix E).

First, we compared the computational speed on a CPU and GPU by comparing the integration time-steps per second (see Table 1). The $\vec{v}$-Net as well as the $\vec{a}$-Net are significantly faster than PhiFlow (11x on CPU and 40x on GPU) as they do not rely on an iterative conjugate gradient solver but instead use a single forward pass through a convolutional neural network that can be easily parallelized on a GPU. To provide a fair comparison on $L_d$, we set the velocity field at the boundaries equal to $\vec{v}_d$. This enables us to compute $L_d$ for the $\vec{a}$-Net architecture on the domain boundaries which would otherwise have zero divergence everywhere. This way, $L_d$ can be interpreted as a metric on how well the orthogonal components of the Dirichlet boundary conditions are met (i.e. no flow leaks through the boundaries). For $dt = 4$, we outperformed Phiflow by several orders of magnitude. For both, $L_d$ and $L_p$, the $\vec{a}$-Net architecture significantly outperformed the more naive $\vec{v}$-Net approach.

Furthermore, we investigated stability by evaluating the evolution of $L_p$ and $L_d$ for the $\vec{a}$-Net over time (see Figure 4). As the fluid state is initialized with $a_z = 0$ and $p = 0$, the $\vec{a}$-Net has to perform a cold-start which is the reason for high $L_p$ and $L_d$ during the first circa 70 steps. Afterwards, the $\vec{a}$-Net continues an accurate and stable fluid simulation.

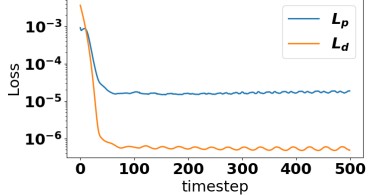

| Method | CPU [TPS] | GPU [TPS] | $L_d$ | $L_p$ |
|---|---|---|---|---|
| PhiFlow | 7 | - | 6.2e-4 | - |
| $\vec{v}$-Net (ours) | **82** | **311** | 8.66e-7 | 4.87e-5 |
| $\vec{a}$-Net (ours) | **82** | **311** | **5.44e-7** | **1.56e-5** |

**Table 1:** Quantitative comparison of timesteps per second (TPS) on CPU / GPU as well as divergence loss and momentum loss for differentiable fluid solvers on a 100x100 grid for viscosity $\mu = 0.1$, density $\rho = 4$ and timesteps of size $dt = 4$.

**Figure 4:** Long term stability of fluid simulations performed by the $\vec{a}$-Net

### 4.3 Optimal Control of Vortex Shedding Frequency

In this section, we present a proof-of-concept experiment that aims at controlling the shedding frequency of a Kármán vortex street behind an obstacle by changing the flow speed (see Figure 5a). To this end, we exploit our previously trained differentiable fluid models.

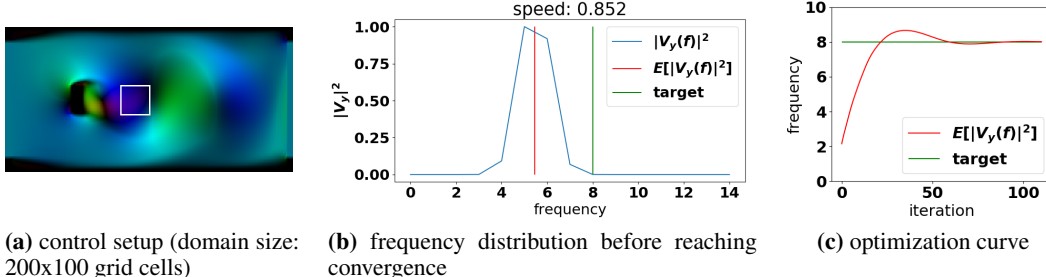

**(a)** control setup (domain size: 200x100 grid cells)

**(b)** frequency distribution before reaching convergence

**(c)** optimization curve

**Figure 5:** The frequency of vortex streets can be controlled using our differentiable fluid models.

First, we measure the y-component of the velocity field $v_y(t)$ behind an obstacle (see white box in Figure 5a) over 200 time steps. Then, we compute the frequency spectrum $V_y(f)$ of $v_y(t)$ using the fast Fourier transform (see Figure 5b). Now, we want to adjust the inflow / outflow boundary conditions in $\vec{v}_d$ such that $E[|V_y(f)|^2] = \hat{f}$. Here, $\hat{f}$ is the target frequency. To optimize $\vec{v}_d$, we define a loss function $L = (E[|V_y(f)|^2] - \hat{f})^2$ and compute the gradients $\frac{\partial L}{\partial \vec{v}_d}$ with backpropagation through time. This is possible since all parts of the loss function including the fluid simulation that is performed by our trained neural fluid model as well as the fast Fourier transform are differentiable. Computing the gradients with a standard automatic differentiation library (Pytorch) took 3.5 seconds for all 200 time steps on our 200x100 domain setup. This is considerably faster than the current state-of-the-art differentiable fluid solver by Takahashi et al. (2021) which takes 5.42 seconds for only 30 time steps on a smaller 128x128 grid. The update steps of $\vec{v}_d$ are done using the ADAM-optimizer and converge after approximately 70 iterations (see Figure 5c). We want to emphasize that differentiable fluid simulations are limited to scenarios with low Reynolds numbers as in the presence of turbulences, chaotic behavior will lead to exploding gradients.

## 5 Discussion and Outlook

In this work, we present an unsupervised learning scheme for the incompressible Navier-Stokes equations and introduce a fluid model that uses a vector potential to output divergence-free velocity fields. Qualitative results of our trained fluid models are in good accordance with expectations from fluid dynamics for a wide range of Reynolds numbers and generalize to unknown fluid domains. Quantitative assessment showed superior performance in terms of accuracy and speed compared to Phiflow and an ablation study that directly predicts the velocity field. We present a real-time demo and demonstrate how differentiability can be used in a proof-of-concept fluid control scenario. We believe that our fluid models can significantly speed up more sophisticated fluid control pipelines such as described by Holl et al. (2020).

First experiments of extending this approach to 3D deliver encouraging results and are topic of future research. Furthermore, on top of Dirichlet boundary conditions, Neumann boundary conditions and multi-phase domains could be incorporated in future fluid models as well.

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

# A  PHYSICS-CONSTRAINED LOSS ON A MAC GRID

As mentioned in Section 3.3 of the paper, our method relies on a staggered marker-and-cell grid representation for the vector potential as well as the velocity and pressure fields. In the following, we provide further details on how to apply this representation to learn incompressible fluid dynamics.

To calculate the velocity field $\vec{v} = \nabla \times \vec{a}$ of a vector potential $\vec{a}$ on a MAC grid in 2D, we have to compute the curl as follows:

$$
\begin{aligned}
(v_x)_{i,j} &= (a_z)_{i+1,j} - (a_z)_{i,j} \\
(v_y)_{i,j} &= (a_z)_{i,j} - (a_z)_{i,j+1}
\end{aligned}
\tag{14}
$$

If this vector potential is inserted into the divergence operator on a MAC grid, we can show that $\nabla \cdot \vec{v}_{i,j} = 0$ is indeed fulfilled:

$$
\nabla \cdot \vec{v}_{i,j} = (v_x)_{i,j+1} - (v_x)_{i,j} + (v_y)_{i+1,j} - (v_y)_{i,j}
\tag{15}
$$

$$
\begin{aligned}
= \; &((a_z)_{i+1,j+1} - (a_z)_{i,j+1}) - ((a_z)_{i+1,j} - (a_z)_{i,j}) \\
&+ ((a_z)_{i+1,j} - (a_z)_{i+1,j+1}) - ((a_z)_{i,j} - (a_z)_{i,j+1})
\end{aligned}
\tag{16}
$$

$$
= 0
\tag{17}
$$

Thus, for the $\vec{a}$-Net, the incompressibility equation is automatically fulfilled and no further training on the divergence loss $L_d$ is required. However, for the $\vec{v}$-Net, the residuals of the divergence are still of importance:

$$
(R_d)_{i,j}^{t+dt} = \nabla \cdot \vec{v}_{i,j}^{t+dt} (= 0 \text{ for } \vec{a}\text{-Net})
\tag{18}
$$

The residuals of the momentum equation in $x$-direction can be computed as follows:

$$
\begin{aligned}
(R_{p_x})_{i,j}^{t+dt} = \rho \Bigg( &\frac{(v_x)_{i,j}^{t+dt} - (v_x)_{i,j}^{t}}{dt} + (v_x)_{i,j}^{t'} \cdot \frac{(v_x)_{i,j+1}^{t'} - (v_x)_{i,j-1}^{t'}}{2} \\
&+ \frac{\frac{(v_y)_{i,j-1}^{t'} + (v_y)_{i,j}^{t'}}{2} \cdot \left((v_x)_{i,j}^{t'} - (v_x)_{i-1,j}^{t'}\right) + \frac{(v_y)_{i+1,j-1}^{t'} + (v_y)_{i+1,j}^{t'}}{2} \cdot \left((v_x)_{i+1,j}^{t'} - (v_x)_{i,j}^{t'}\right)}{2} \Bigg) \\
&+ \left(p_{i,j}^{t+dt} - p_{i,j-1}^{t+dt}\right) - \mu \cdot \Delta(v_x)_{i,j}^{t'}
\end{aligned}
\tag{19}
$$

Here, we use the following isotropic Laplace operator:

$$
\begin{aligned}
\Delta s_{i,j} = \frac{1}{4}( &1 * s_{i-1,j-1} + 2 * s_{i-1,j} + 1 * s_{i-1,j+1} \\
&+ 2 * s_{i,j-1} - 12 * s_{i,j} + 2 * s_{i,j+1} \\
&+ 1 * s_{i+1,j-1} + 2 * s_{i+1,j} + 1 * s_{i+1,j+1})
\end{aligned}
\tag{20}
$$

The derivation of the advection term for $R_{p_x}$ is a bit more complex since on a MAC grid, $v_x$ and $v_y$ are displaced by half a pixel in $x$-direction and $y$-direction. To obtain the residuals of the momentum equation in $y$-direction, $(R_{p_y})_{i,j}$, one has to take $(R_{p_x})_{i,j}$ and swap $x$ and $y$ and the indices respectively.

Now, the discretized loss terms can be written as follows:

$$
L_d^{t+dt} = \sum_{i,j} \Omega_{i,j}^{t+dt} ((R_d)_{i,j}^{t+dt})^2
\tag{21}
$$

$$
L_p^{t+dt} = \sum_{i,j} \Omega_{i,j}^{t+dt} \left(((R_{p_x})_{i,j}^{t+dt})^2 + ((R_{p_y})_{i,j}^{t+dt})^2\right)
\tag{22}
$$

$$
L_b^{t+dt} = \sum_{i,j} \partial\Omega_{i,j}^{t+dt} \left\| \vec{v}_d^{t+dt} - \vec{v}^{t+dt} \right\|^2
\tag{23}
$$

Note, that all mentioned operations can be efficiently implemented with convolutions. To obtain the final velocities on a square grid, we project the velocity fields of the MAC grid back onto the $\vec{a}$-grid using linear interpolation:

$$\vec{v} = \frac{1}{2} \begin{pmatrix} (v_x)_{i-1,j} + (v_x)_{i,j} \\ (v_y)_{i,j-1} + (v_y)_{i,j} \end{pmatrix} \tag{24}$$

## B    NETWORK ARCHITECTURE

Our fluid model is based on the U-Net architecture (Ronneberger et al. (2015)) with fewer channels (see Figure 6). As the pressure field and vector potential can have an arbitrary offset, we always normalize the mean of the pressure ($\Delta p$) and vector potential ($\Delta a_z$) to 0 to keep these fields well-defined and prevent drifting offset values.

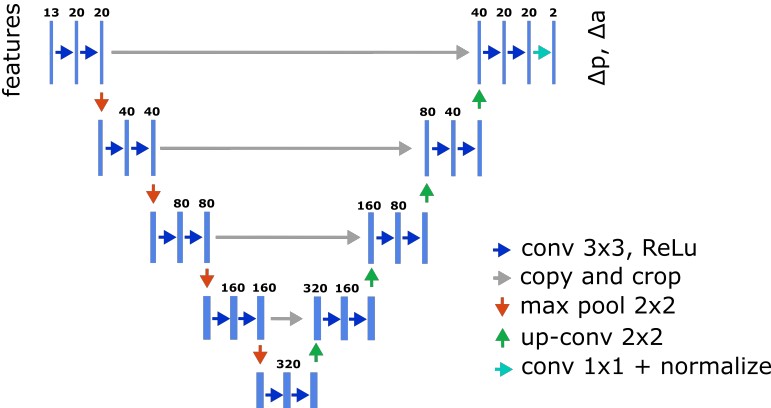

**Figure 6:** U-Net architecture with fewer channels.

## C    EXAMPLES OF TRAINING DOMAINS

The domains we used for training consist of $100 \times 300$ grids. We used 3 different randomized domains as exemplary depicted in Figure 7. First, we have boxes with randomized height and width that float on randomized paths inspired by Brownian motion in a pipe with randomized flow speed. Second, we have the same setup but replaced the boxes by cylinders with randomized radii and angular velocities in order to learn the Magnus effect. Finally, we have a folded pipe system with randomized flow speed, that is randomly flipped along the $x$-axis.

## D    FURTHER EXAMPLES OF GENERALIZATION

Note that the network was only trained on simple domain geometries as presented in appendix C. Still, as can be seen in Figure 8, the network is capable of generalizing to far more complicated domain geometries (e.g. shark, car). Figure 8c shows that it can generalize to multiple objects in the scene, although the training set contained at most one object per scene. And Figure 8d shows that we can alter the outer boundary conditions as well. For real-time simulations, please have a look at our source code and the supplementary video.

## E    QUANTITATIVE ANALYSIS: THE BENCHMARK PROBLEM

Figure 9 shows the domain $\Omega$ and $v_d$ on a $100 \times 100$ grid which was used as the benchmark problem for quantitative analysis. The flow speed for the inlet and outlet was set to 0.5. The timestep of the integrator was set to $dt = 4$ and the viscosity and fluid density were set to $\mu = 0.1$ and $\rho = 4$ respectively.

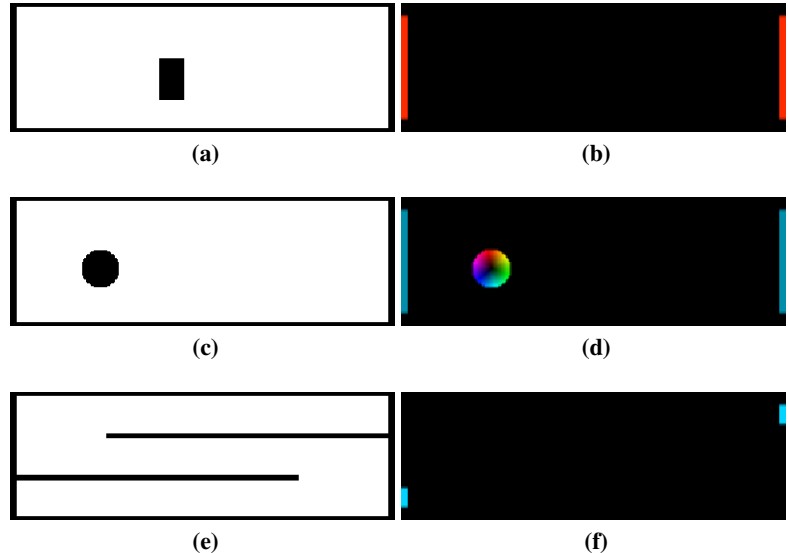

**Figure 7:** The left column shows $\Omega$ (in white) / $\partial\Omega$ (in black) and the right column shows $\vec{v}_d$ for three examples of training domains. (Colors indicate the direction and magnitude of $\vec{v}_d$ as depicted in Figure 9a)

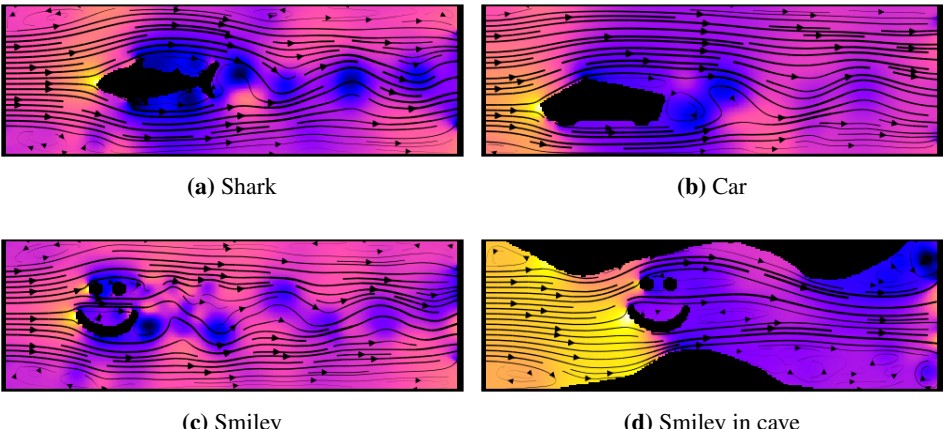

**(a)** Shark      **(b)** Car

**(c)** Smiley      **(d)** Smiley in cave

**Figure 8:** Our models generalize to various domain geometries, although being trained only on simple shapes (see Figure 7)

## F QUALITATIVE COMPARISON OF $\vec{a}$-NET AND $\vec{v}$-NET

We give a qualitative example to show the benefits of using a vector potential. Figure 10 demonstrates that the $\vec{a}$-Net finds plausible solutions for a folded pipe domain while the $\vec{v}$-Net looses most of the flow in the center of the domain. This is in good accordance with quantitative results shown in section 1. The folded pipe domain is particularly difficult to learn as the flow field contains long range dependencies to the inlet and outlet (as shown in the bottom row in Figure 7).

## G TRAINING WITHOUT RESETTING ENVIRONMENTS

We performed an ablation study to investigate what happens if we do not reset old environments from time to time and, thus, do not continuously present the fluid model with cold starts during training. Figure 11 shows that in this case, large error spikes appear in the validation curve. These error spikes appear since the model has troubles to perform a cold start as can be seen in Figure 11b: compared

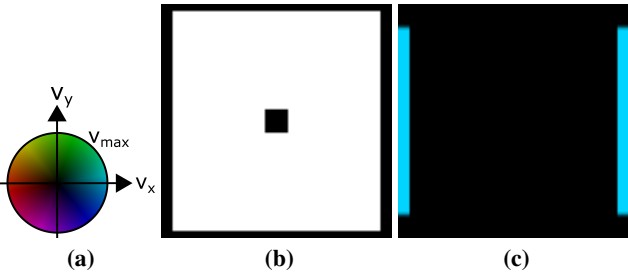

**Figure 9:** a) shows legend for $\vec{v}_d$; b) shows $\Omega$ (in white) / $\partial\Omega$ (in black) for the benchmark problem; c) shows $\vec{v}_d$ for the benchmark problem. (Colors indicate the direction of $\vec{v}_d$ as depicted in a)

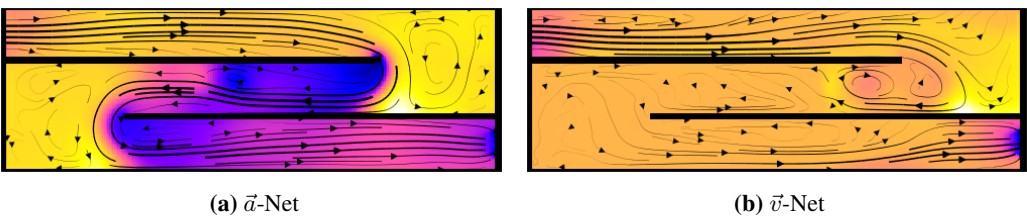

**(a)** $\vec{a}$-Net

**(b)** $\vec{v}$-Net

**Figure 10:** Qualitative comparison of $\vec{a}$-Net and $\vec{v}$-Net in a folded pipe domain

to a properly trained model (see Figure 4) the model takes longer to perform a cold start (ca 100 steps) and converges to a solution with high $L_p$- and $L_d$- losses. By resetting the environments from time to time during training, we can prevent these error spikes as shown in Figure 11c.

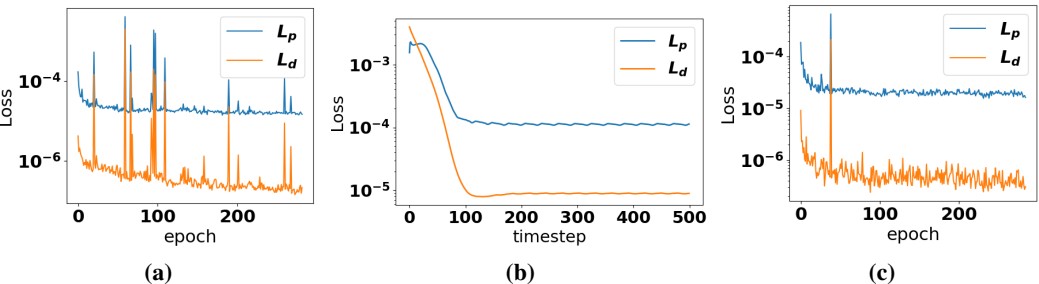

**(a)**

**(b)**

**(c)**

**Figure 11:** a) ablation study without resetting environments: validation curve shows large error spikes during training; b) error spike: the fluid model takes longer to perform a cold start and converges to a solution with high losses; c) original training with resetting environments: validation curve is stable

