# OpenReview forum: "Learning Incompressible Fluid Dynamics from Scratch - Towards Fast, Differentiable Fluid Models that Generalize"
_ICLR.cc/2021/Conference — ICLR 2021 Spotlight_

### Official Review · AnonReviewer1 · 2020-10-20
**A nice paper that might be improved with more information & simple baselines**

**Rating:** 7
**Confidence:** 3

**Review:**

### Summary of my understanding

The authors propose an unsupervised (or rather, I would say self-supervised) method to build a simulator of incompressible fluid flows based on neural networks. Their neural network is given the pressure and velocity (or its potential) fields at time $t$ and is trained so that it outputs the fields at time $t+dt$ with the "physics-constrained loss functions." The proposed loss functions are designed to make the outputs of the neural net fulfill the Navier-Stokes equation and boundary conditions. They train the neural net on randomly-generated domains and boundary conditions. They show that the learned model outputs qualitatively plausible flows, even if the domain is not exactly handled in the training phase.

### Evaluation

The paper is easy to follow. The proposed method is technically reasonable. The related work section is comprehensive. While the technical novelty of each component of the proposed method looks moderate, the overall framework would be valuable as a fast differentiable fluid dynamics solver, which the authors claim. The experiment section is convincing to some extent, but it lacks comparison to classical solvers (i.e., somewhat inherently-faithful simulations), which limits our capability to assess the soundness of the results in Section 4.1 and Appendix D. Also, the paper lacks information on the range of hyperparameter search.

### Questions

[Q1]
How did you create the randomized boundary conditions $(v_d)_k^0$? I could not find it in the paper.

[Q2]
The training strategy described in Section 3.7, especially the pool's update and random renew, seems essential to the performance of the proposed method. Did you performed some ablation study in this regard, or do you have some notes on what happened if the presented strategy was not adopted partially? Such a description would help a reader's understanding much.

[Q3]
Can you provide the range of hyperparameter search and information on how intensively the search was done? Such information is important for assessing training computational cost.

---

> ### Author Response · Authors · 2020-11-15
> **Reply**
>
> Thanks a lot for your kind and constructive review!
>
> Regarding the comment on lacking comparisons to classical solvers, we want to clarify our motivation for choosing PhiFlow by Holl et al.: Phiflow is a well-documented, recent open-source library for fluid-simulations and we consider it to be conceptually relevant, as it provides - like our trained fluid models - differentiable fluid-simulations based on a MAC-grid data structure.
>
> In the following, we provide answers to the specific questions:
>
> Q1: Figure 7 (in Appendix) shows examples of randomized domains and boundary conditions that were used during training. As you can see, these domains are really simple and still the network is capable of generalizing to much more complicated domains (see e.g. Fig. 8). As mentioned in our reply for Reviewer 2, possible reasons could be:
> 1. during training, the network gets confronted with basically an infinite number of different flow-fields and randomized domain configurations because the training pool gets updated at every training step. This prevents the network from over-fitting.
> 2. the dynamics of a fluid-particle is mostly determined by its local neighborhood / surrounding particles. This means, the update step for a certain cell on the MAC grid is mostly determined by very close / neighboring MAC-grid cells. Since more complicated shapes can be seen locally as a composition of basic shapes (e.g. the fin of the shark can be locally regarded as a triangle), it suffices to train on basic shapes that provide the network with enough examples to generalize to more complicated shapes.
>
> Q2: Indeed, updating the pool is a very crucial step. Otherwise, the fluid states would stay 0 all the time and our method would not work as there would not be any realistic fluid states present in the pool to train on the network. The pool’s random renew was chosen in order to provide the network with a greater variety of randomized domains and also learn cold-starts. We didn’t perform an ablation study in this regard yet but if demanded I think we can definitively add such a study in the appendix until 24. of November.
>
> Q3: Actually, there are not many hyperparameters to optimize so we did it by hand. The learning rate (=0.001) in Adam was set to a pretty standard value and the batch size (=100) was chosen such that everything still fits on one GPU. The weights of beta (=1) and gamma (=20) were chosen such that the corresponding loss terms converged to reasonable values and no artifacts (such as e.g. flow leaking through boundaries) appear. If you want to train your own fluid model using our provided code you should be fine with our default values. Regarding computational costs for training: Training each network took us about 1 day on a NVidia GeForce RTX 2080 Ti.
>
> We hope, we were able to answer your questions. If there are further remarks or questions left, feel free to write us again!

---

### Official Review · AnonReviewer3 · 2020-10-27
**Interesting direction with open issues from previous submission**

**Rating:** 7
**Confidence:** 5

**Review:**

This paper proposes to learn the dynamics of an incompressible fluid via a physics informed loss formulation using an unsupervised training framework. It employs a custom solver that is executed at training time to learn a Navier-Stokes residual with a incompressible (curl of a stream function) formulation. This setup is demonstrated for two dimensional karman vortex streets, and a control example for the magnus effect.

The paper definitely targets an interesting and relevant direction for machine learning, but it is a (fairly identical) resubmission of a paper that was rejected at NeurIPS. It is of course possible to resubmit papers, but unfortunately in this case it was a fairly clear rejection, and while I cannot compare versions side by side, I think the submitted content is largely the same. As both NeurIPS and ICLR are currently mostly on par in terms of aims, content, and expectations for accepted submissions, I cannot recommend accepting this paper in its current form to ICLR.

For the previous submission, I believe the following points were the most important issues, as raised in the reviews: Most importantly, the benefits of the proposed method over existing ones (ie supervised from regular solver) are not made clear enough. Then, there is the insufficient quantitative evaluation, that the stream function formulation does not properly extend to 3D, and that the direction was not fully clear (performance vs control applications).

I'm aware that it must be frustrating for authors to be rejected for very similar reasons, but it is likewise not nice for reviewers to repeatedly look at papers searching for differences, and discovering that authors have not taken comments from previous submissions into account. I would be curious to hear in the rebuttal how or whether the authors have updated their submission after the NeurIPS reviews. As already mentioned in the previous cycle: there is merit to the direction of the paper, but I think it is important to make a clear step forward for one or more of the issues raised with the submission (as outlined in the previous paragraph). With such extensions, there should be the potential for an improved evaluation. However, as it stands I don't think this submission is suitable for ICLR.

A minor note, but the title seems somewhat inapproprate to me - physics-informed typically implies that derivatives are computed via autodiff from a network representing the solution. Here, the authors instead discretize the solution on a grid, while a network yields the solution for the next step, and the physical model is evaluated with FDs on the grid. "Physics-constrained" (as in the section title) would be a better choice for the title.

---

> ### Author Response · Authors · 2020-11-15
> **Reply**
>
> We are very surprised regarding the reviewer's feedback.
> It is true that we already submitted this work to NeurIPS and got rejected (scores: 4 (confidence 2),5 (confidence 4),5 (confidence 4)). But since then, we put a lot of time and effort into improving the paper:
> 1. We rewrote to a large extend the related work part (section 2), as the reviewers pointed out that we should remove less related parts.
> 2. We also rewrote to a large extend the methods part (section 3). There were some major misunderstandings in the reviews at NeurIPS: e.g. one reviewer believed, we would use some kind of differentiable solver during training and another one believed that computing the loss function would have similar complexity compared to solving the Navier Stokes equations. This is clearly not the case as we clarified in section 3.5: computing the loss can be done very efficiently in O(N) (where N~number of grid points). Solving a system of N equations would be a lot more expensive!
> 3. We added a stability analysis over time (see Figure 4) to strengthen our quantitative analysis.
> 4. We completely rewrote the control task section (Section 4.3) and moved it from the appendix into the main body of the paper. This was requested by a reviewer at NeurIPS and hopefully clarifies this section and the direction of the paper.
> 5. We tidied up and published the code (and already got positive feedback from researchers). Now, it is easily possible to check the code and reproduce our results.
> 6. We did a lot of cosmetic improvements including changing the title and enhancing visual style and clarity of Figure 1 a) & b).
> 7. We added further generalization examples (see e.g. Figure 3b, 8d, end of movie)
> 8. There is absolutely no evidence for the argument that stream functions do not extend to 3D. In fact, there are multiple deep learning based algorithms that use stream functions / vector potentials in 3D (e.g. Kim et Al. (2019), Mohan et Al. (2020), Raissi et Al. (2019)) and first experiments from our side in this direction are promising as well.
>
> For this reason, accusing us of not taking reviewer comments into account seems inadequate.
>
> Furthermore, this review claims we would “employ a custom solver that is executed at training time”. We took great effort to resolve this misunderstanding at NeurIPS (see point 2). In addition, the review claims we would demonstrate “a control example for the magnus effect” when we actually demonstrate an example to control the shedding frequency of a von Karman vortex street. These are fairly basic points in the paper that should be clear after reading the paper.
>
> Finally, we want to emphasize that we are always very happy about constructive feedback that helps to improve our submissions!

---

### Official Review · AnonReviewer4 · 2020-10-28
**Good paper. But an absence of baselines for comparison.**

**Rating:** 7
**Confidence:** 5

**Review:**

The paper is generally a good contribution especially in the field of machine learning for physics. However, I do have some questions about the novelty of this work and what makes it different from other physics informed approaches. Below are the pros and cons and the I list out a set of comments and additional results that I believe would make the contribution of the paper much more clear.

Pros:
-- A physics informed neural architecture that optimizes the governing PDEs to perform forward integration of the system. Generally a great approach since it does not require high-res numerical simulations
-- Generalizes to new geometries

Cons
-- I might be missing something, but I generally feel that there is a lack of novelty. Approach is very similar to certain papers I point out in the comments. That however, does not take anything away from the approach proposed and it is still a good contribution once certain concerns (in the comments) are resolved
-- The authors claim that their network generalizes but I do not see an explanation for it. Even an intuitive explanation would be good.

1) The introduction section of the paper does talk about different approaches to physics informed neural networks especially the Raissi et al., papers and mention that Raissi's approach (https://arxiv.org/pdf/1711.10566.pdf and https://arxiv.org/abs/1711.10561) does not generalize to new domains. Can the authors explain why they believe that their network would automatically generalize to new geometries? Does the introduction of $\Omega$ and $\partial \Omega$ in their input features make their framework generalizable?

2) The use of vector potentials to ensure divergence free by construction inside neural architectures using fixed convolution operations has been shown before By Mohan et al., (https://arxiv.org/pdf/2002.00021.pdf). The authors should cite this paper.

3) In Figure 2, I'm concerned about how well the network satisfies BCs. If possible, I would like to see a plot showing velocity with time during inference near the edges of the obstacle.

4) There is a complete absence of baselines. Sure, the a-net outperforms v-net when it comes to the loss. However, the loss term is $10^{-3}$ for v-net and $10^{-5}$ for a-net. Both are pretty low. How much improvement in inference would one expect?  It is evident that constraining divergence by construction definitely helps in reducing the loss and this has been shown in https://arxiv.org/pdf/2002.00021.pdf.

5) I would like to see a comparison between the a-net prediction and a numerical simulation and possibly a comparison with general PINNs (https://arxiv.org/pdf/1711.10566.pdf)  for the rectangular obstacle. It should clearly show that their approach is more accurate in terms of RMSE error between a-net and numerical simulation as compared to PINN and numerical simulation. What would be more interesting to see is how well the authors satisfy BCs as compared to PINN. Moreover according to the authors PINNs would not generalize to the airfoil shaped obstacle while their network would . This would clearly show an advantage of the proposed mechanism. Without a baseline however, there is not much to prove that this is a better approach.

6)  The problem solved in this paper is a very simple problem. Even numerical solvers are not expensive when it comes to solving this problem. This physics informed approach is very promising because it can reduce the computational cost immensely (during inference). Would it be possible for the authors to show this for a *real* turbulent flow, e.g. the system considered in the Mohan et al., paper (https://arxiv.org/pdf/2002.00021.pdf) or the Kolmogorov system as shown in this paper (https://advances.sciencemag.org/content/advances/3/9/e1701533.full.pdf) or even the 2D Kraichnan system shown in https://arxiv.org/abs/1808.02983. Unless these approaches generalize to fully turbulent flow, the use for such architectures are limited for real applications

This is still an important contribution in the field of deep learning for computational physics, specifically in fluid dynamics.

---

> ### Author Response · Authors · 2020-11-15
> **Reply**
>
> Thanks for the kind and constructive feedback and we are very happy to hear that this work makes an important contribution in the field of deep learning for computational physics / fluid dynamics!
>
> We will take great effort to handle the mentioned suggestions in the updated version of the paper.
>
> Major contributions and benefits of our approach are the following:
> 1. Our approach does not require any data from fluid simulations beforehand. This greatly reduces computational and memory costs for the dataset.
> 2. Our approach generalizes well to previously unseen domains.
> 3. By taking the full incompressible Navier-Stokes equations and dirichlet boundary conditions into account, our approach allows to capture physical phenomena like the Magnus effect within the simulation.
> 4. Our approach allows for optimal control by propagating gradients through the learned fluid simulation.
>
> Unfortunately, there is no theoretical proof about the networks generalization capabilities but we can certainly provide some further intuition. Possible reasons might be:
> 1. during training, the network gets confronted with basically an infinite number of different flow-fields and randomized domain configurations because the training pool gets updated at every training step. This prevents the network from over-fitting.
> 2. the dynamics of a fluid-particle is mostly determined by its local neighborhood / surrounding particles. This means, the update step for a certain cell on the MAC grid is mostly determined by very close / neighboring MAC-grid cells. Since more complicated shapes can be seen locally as a composition of basic shapes (e.g. the fin of the shark can be locally regarded as a triangle), it suffices to train on basic shapes that provide the network with enough examples to generalize to more complicated shapes.
>
> We are happy to improve our paper by clarifying the reviewer’s specific questions:
> 1. Yes, we definitively need to input information about the domain / domain boundaries to the network in order to generalize to new domain boundaries. (Otherwise, the network would have no chance to “see” and adapt to new domains.)
> 2. Thanks for the reference, we’ll include it in the updated version.
> 3. This is an interesting point. So far, we didn’t look into the velocity components close to the domain boundaries since in general we also allow for moving boundaries with v_d != 0. Furthermore, for high Reynoldsnumbers, the boundary layer can be very small and the parallel component of the velocity field with respect to the domain boundary can significantly deviate from 0 - even at a distance of only 1 grid cell. However, the divergence loss (see Table 1 and Figure 4) also captures the divergence at the domain boundaries and would return big values if the flow would “leak” through a boundary. This at least ensures that the orthogonal component of the velocity field with respect to the domain boundary fullfills the no-slip boundary conditions. Does this answer your question?
> 4. Yes, constraining divergence by construction definitively helps during inference - not only quantitatively (as shown in Table 1) but also qualitatively. We will include a qualitative example that clearly shows the benefit of the a-Net over the v-Net in a pipe-scenario in the appendix.
> 5. Yes, PINNs as introduced by Raissi et al. do not generalize to new domain geometries by design. (For example, they do not have an extra input to obtain information about the domain geometry). Furthermore, they generate continuous solutions for the Navier Stokes equations making it hard to compare against our discrete MAC-grid solutions. That’s why we decided to rather compare our method against Phiflow.
> 6. Turbulent flow: we looked at a simple example with 0 viscosity (see Figure 2d) and observed some chaotic / turbulent fluid behavior. However, this case required a regularization term on the pressure gradient during training and shows less chaotic behavior as for example the suggested paper by Mohan et Al.. In our case, simulating fully turbulent flow was not the main goal, since it doesn’t allow for gradient propagation over reasonably long time horizons. This is because the chaotic behavior of turbulent flows would lead to exploding gradients (see e.g. the butterfly effect). Nevertheless, this is a very important direction for future research and we’ll try to add results with more turbulent flow in our supplementary video.
>
> We hope, we were able to answer your questions. If there are further remarks or questions left, feel free to write us again!

---

> > ### Comment · AnonReviewer4 · 2020-11-24
> > **Response**
> >
> > The authors have sufficiently answered my questions. It is also evident that most of the issues that I had raised were already considered by the authors, especially regarding the boundary conditions. I also appreciate the authors adding the new analysis and conducting experiments on an extra test case. I feel this paper is well written and (from my previous comment) an important contribution to building deep learning driven physics models, especially for complex systems. I have changed my rating to "Good paper, accept".

---

### Official Review · AnonReviewer2 · 2020-10-29

**Rating:** 7
**Confidence:** 3

**Review:**

- Summary

This paper presents a "physics-informed" deep learning model of fluid dynamics. The underlying deep learning architecture employed is a somewhat standard u-net, but one of the proposed method's distinguishing features is that it enforces its adherence to physical behavior at its loss terms, by penalizing predictions that are not incompressible or do not conserve momentum. Notably, this approach allows it to be trained unsupervisedly, without requiring the generation of ground-truth simulations.

- Pros

The enforcement of the physical behavior as a core feature of the architecture (e.g., through the "a" vector and the conservation losses) makes the network to generate stable simulations.

This also brings with it the interesting benefit of being able to learn without the need for ground truth data, lowering the cost of training the model.

The proposed model performs favorably, both in terms of speed and error, to phiflow, the current "reference" differentiable fluid simulator. Since the proposed method is still deep learning based, it is also differentiable.



- Cons

Most of the experiments serve only to validate the model's accuracy and its ability to learn a proper fluid simulation.
The only practical application demonstrated is in a simple control task, which is not explored too deeply.
Phiflow (Holl et al., 2020), for example, performs more extensive and diverse evaluation of control settings.


- Reasons for score

Overall, given the "pros" described above, notably the distinct loss formulation that allows the model to learn unsupervisedly to perform efficient, differentiable fluid simulations, I recommend this paper for acceptance.


- Additional comments

Is training improved if actual simulations are used for data, instead of "cold starts"?
If so how do these compare with training with cold starts (as presented in the paper), both in terms of training speed and in terms of final (test) results?
Does starting from more realistic starting points fix the issue of the initial error "spike" in Fig 4?

---

> ### Author Response · Authors · 2020-11-15
> **Reply**
>
> Thanks a lot for the kind and constructive review!
>
> Indeed, the paper “Learning to Control PDEs with Differentiable Physics” by Holl et al. performed a very thorough evaluation of control settings. However, in order to train the control network, their method relies on a “traditional” differentiable fluid-solver (Phiflow) to obtain a differentiable loss function. In this work, we didn’t focus primarily on learning a “fluid-controller”, but instead, we mainly focus on learning a “fluid-solver” that provides us with fast, accurate and differentiable fluid simulations. We believe that the training-pipeline of Holl et al. could be significantly sped up by replacing their “traditional” Phiflow-solver with our learned fluid-solver and this is definitively a direction to continue research on in the future.
>
> So far, we didn’t test if training with actual simulated data improves training. We guess, it might speed up training a little bit at the beginning, since at the beginning, our training pool doesn’t contain realistic fluid states (they are basically all set to 0). However, generating simulated data for a very large number of domains and timesteps in the first place is computationally and memory-wise very expensive.
> In contrast, progressively enriching the dataset with more and more realistic fluid states of randomized domains during training provides the network in the long run with a much larger variety of different fluid states at basically no additional computational / memory costs and yields very stable fluid-simulations that generalize to previously unseen domains.
>
> Cold-starts at the beginning of a simulation are a common problem and also lead to higher errors in conventional fluid-solvers (we observed the same phenomenon in our tests with the phiflow framework as well). The problem stems from the sudden change of fluid velocity at the beginning of the simulation.
> So to answer your last question: yes, starting from a “warmed-up” (more realistic) fluid state definitively fixes the issue of the initial error spike.
>
> We hope we could answer your questions. If there are further remarks or questions left, feel free to write us again!

---

> > ### Comment · AnonReviewer2 · 2020-11-24
> > **Response**
> >
> > Thanks for the response. The answers presented here clarify my previous questions. I am fine with idea that a fluid controller might be left for future work, as I believe this work already contains enough value as it is so as to be of interest to a wide ICLR audience. The additions to the paper in response to the comments by other reviewers are also welcome. I will maintain my (already positive) evaluation of the paper.

---

### Author Response · Authors · 2020-11-23
**Updated Submission**

Dear Reviewers,

thanks a lot for your valuable feedback!

We took great effort to include as many comments as possible into our
updated submission:

 1. We changed the title to “Learning Incompressible Fluid Dynamics from
    Scratch - Towards Fast, Differentiable Fluid Models that
    Generalize”. We believe this title is more specific and places a
    stronger focus on our core contributions: learning fast,
    differentiable fluid models that generalize without fluid simulation
    data.
 2. We added a qualitative comparison between the a-Net and the v-Net in
    Appendix F to show the benefit of using a vector potential over
    directly predicting the velocity field.
 3. We added further results of non-deterministic wake dynamics in case
    of high Reynolds numbers in our supplementary video.
 4. We additionally discussed the mentioned reference by Mohan et al. in
    our related work section.
 5. We added intuitive explanations on why our approach is able to
    generalize in Section 4.1.
 6. We added an ablation study to demonstrate the effect of not
    resetting old environments from time to time during training in
    Appendix G.
 7. We put a lot of effort into revising formulations to improve the
    readability. Furthermore, we added an animation of the recurrent
    fluid model in the video.

We hope, that the changes summarized above and the respective detailed
discussions w.r.t. the individual reviewer responses clarify your
concerns. If there are further remarks left regarding our (updated)
submission, feel free to contact us!

Best regards, anonymous authors

---

### Decision · Program_Chairs · 2021-01-07
**Final Decision**

**Decision:**

Accept (Spotlight)

**Comment:**

The paper introduces a learning framework for solving incompressible Navier-Stokes fluid using a physics informed loss formulation. The PDE is solved on a grid, and the model, implemented via convolutions and a U-Net, is trained to minimize the NS residual. The model is trained on a variety of randomized contexts, in a way that allows training to explore a large number of configurations. The paper presents original contributions compared to previous Physics informed framework (discrete formulation, conditioning on the domain conditions, …). All the reviewers agree that the detailed rebuttal provides answers to their questions and that the contribution is significant, they all have a positive assessment of the paper.